# Investigation of a Direct Interaction between miR4749 and the Tumor Suppressor p53 by Fluorescence, FRET and Molecular Modeling

**DOI:** 10.3390/biom10020346

**Published:** 2020-02-22

**Authors:** Anna Rita Bizzarri, Salvatore Cannistraro

**Affiliations:** Biophysics and Nanoscience Centre, DEB, Università della Tuscia, 01100 Viterbo, Italy; bizzarri@unitus.it

**Keywords:** miR4749, p53, fluorescence quenching, FRET, computational docking, oncomiR

## Abstract

The interactions between the DNA binding domain (DBD) of the tumor suppressor p53 and miR4749, characterized by a high sequence similarity with the DNA Response Element (RE) of p53, was investigated by fluorescence spectroscopy combined with computational modeling and docking. Fluorescence quenching experiments witnessed the formation of a specific complex between DBD and miR4749 with an affinity of about 10^5^ M. Förster Resonance Energy Transfer (FRET) allowed us to measure a distance of 3.9 ± 0.3 nm, between the lone tryptophan of DBD and an acceptor dye suitably bound to miR4749. Such information, combined with a computational modeling approach, allowed us to predict possible structures for the DBD-miR4749 complex. A successive docking refinement, complemented with binding free energy calculations, led us to single out a best model for the DBD-miR4749 complex. We found that the interaction of miR4749 involves the DBD L_3_ loop and the H_1_ helix, close to the Zn-finger motif; with this suggesting that miR4749 could directly inhibit the p53 interaction with DNA. These results might inspire new therapeutic strategies finalized to restore the p53 functional activity.

## 1. Introduction

MicroRNAs (miRs) are short non-coding RNAs involved in gene expression controlling many processes at the basis of cell cycle, such as differentiation, development and apoptosis [1,2,3]. Some miRs are overexpressed in many human malignancies where they may act as oncogenes. On such a basis, miRs represent promising biomarkers in clinical diagnosis and prognosis of tumors. Several miRs have been found to be involved in the network of the tumor suppressor p53, which is a master regulator of cell cycle and apoptosis, and mutated in almost half of human cancers [4]. Indeed, miRs can contribute to the regulation of the p53 levels and functions, through a direct control of p53 transcription or, in an indirect way, acting on p53 inhibitors, such as Murine Double Minute 2 (MDM2). On the other hand, p53, and some p53 mutants, are involved in the regulation of miR transcription and maturation by directly interfering with the p53 function [5]. However, a full knowledge of the interaction details and the regulation mechanisms between p53 and miR is far from being reached.

Very recently, it has been demonstrated that miR-23a directly binds to p53, acting on the process that promotes apoptosis [6]. Accordingly, the occurrence of a direct physical association between miRs and p53 could represent a further interesting level of regulation within the miR-p53 network. At the same time, a full comprehension of the miR-p53 interaction could be of foremost importance in cancer biology, also offering an additional way to develop new anticancer strategies. Along this direction, we have recently investigated, in vitro, the interaction between p53 and mature miR-21-3p, which is abnormally expressed in some human tumors and in different cancer cell lines [7]. Such a study, performed by spectroscopic and nanoscopic techniques, has demonstrated the formation of a complex between the DNA binding domain (DBD) of p53 and miR-21-3p, and suggested possible molecular regions involved in the interaction; with this providing clues about new targets for novel drugs in cancer therapy.

Here, we have investigated the interaction between p53 and miR4749, which is involved in rectal cancer and it is characterized by a high sequence similarity with the DNA Response Element (RE) of p53 family members [8]. The interaction between DBD and miR4749 has been investigated by combining fluorescence and computational methods. We have analyzed the fluorescence quenching of the lone solvent-exposed Trp belonging to the DBD portion of p53 as a function of miR4749 concentration, by finding out the formation of a specific complex between DBD and miR4749 with an affinity of about 10^5^ M. To extract structural information on such a complex, we have applied the Förster resonance energy transfer (FRET) method, which is based on the energy transfer between donor and acceptor chromophores, allowing to probe the distance in the 10 and 100 Å range [9]. In our system, the lone Tryptophan of DBD acts as the energy donor, while the acceptor is constituted by an appropriate dye (Atto390), covalently bound to the 5′ end of miR4749. Such an approach has led us to evaluate a distance of 3.9 ± 0.3 nm between the DBD Tryptophan and the dye bound to miR4749. In the perspective to elucidate possible interaction regions, and the related mechanisms, a molecular modeling procedure for the DBD-miR4749 complex has been developed. Modeled structures of miR-4749 have been submitted to a docking procedure with the X-ray structure of DBD by obtaining 50 possible models for the DBD-miR4749 complex. These models have been then screened and refined by both FRET data and free energy calculation to find out a final best model for the complex. We found that the most probable interacting regions between DBD and miR4749, involves its DNA binding site of DBD, with a consequent, possible impairment of the p53 oncosuppressive functions. These results highlight the important role played by a direct interaction between miRs and p53, especially in connection with drug design finalized at restoring the p53 antitumoral function.

## 2. Materials and Methods

### 2.1. Materials

Recombinant human DNA binding domain (DBD) of p53, formed by the 94-300 residues, including a single Tryptophan (Trp 146), was purchased from Genscript (Piscataway, NJ, USA). Single stranded RNA with the sequence of hsa-miR-4749-5p (UGCGGGGACAGGCCAGGGCAUC; 7.1 kDa; hereafter miR4749), alone and labeled with the Atto-390 fluorescent dye at the 5′ end (7.2 kDa; hereafter miR4749Atto390), were purchased from Metabion (Planegg, Germany). Purity of miR4749 and of DBD was verified by HPLC by mass spectroscopy by the producer. A phosphate buffer (PBS) 50 mM at pH 7.4 was prepared using reagents from Sigma–Aldrich Co. (St Louis, MO, USA).

### 2.2. Absorbance and Fluorescence Measurements

Absorption experiments, at 298 K, were performed by Jasco V-550UV/visible spectrophotometer using 1-cm path-length cuvettes and 1-nm bandwidth in the 220–750 nm region, using the PBS buffer as reference. Steady-state fluorescence measurements were carried out by a FluoroMax^®^-4 Spectrofluorometer (Horiba Scientific, Jobin Yvon, France), at 298 K. Fluorescence emission spectra were recorded at 298 K, from 305 to 580 nm with 1 nm increments and an integration time of 0.50 s using an excitation of 295 nm. A 2 nm band-pass width was used in both excitation and emission paths. Emission spectra were acquired in the signal to reference (S/R) mode to minimize random lamp intensity fluctuations, and corrected for Raman contribution from the buffer. Each fluorescence experiment was carried out on ten independently prepared samples. Spectra were analyzed by using the FluorEssence software (Horiba Scientific, Jobin Yvon, France).

Lifetime measurements were performed at 298 K with the time-correlated single photon counting method using FluoroMax^®^-4 Spectrofluorometer (Horiba Scientific, Jobin Yvon, France). The apparatus, operating at a repetition rate of 1 MHz and in reverse mode, was equipped with a pulsed nanosecond LED excitation head emitting at 295 nm (Horiba Scientific, Jobin Yvon, France) with a temporal width lower than 1 ns and a bandwidth of 4 nm. Fluorescence lifetime data were acquired at 345 nm until the peak signal reached 10,000 counts. Data were analyzed by making use of the impulse response function (DAS6 software, Horiba Scientific, Jobin Yvon, France). The intensity fluorescence decay was analyzed as in references [7,10], by evaluating the average fluorescence lifetimes, <τ>, calculated through the expression:<τ>= (∑_i=1_ a_i_ e^−t/τi^)/(∑_i=1_ a_i_), in which two exponential decays were taken into consideration.

### 2.3. Modeling Procedures

The structure of DBD to make the DBD-miR4749 complex was derived from the B chain of DBD in complex with a consensus DNA (1 TUP entry from the protein data bank) [11]. The structure was suitably adjusted to match the DBD portion used in the fluorescence and FRET experiments. In particular, the short 290–300 AA portion was added by using the I-TASSER suite [12].

The structure of DBD is composed by two antiparallel β-sheets of four and five strands, respectively, forming a β-sandwich structure (see Figure 1). The Zn ion is tetrahedrically coordinated to the side-chains of Cys176, His179, Cys238 and Cys242, forming a Zn-finger motif, which is connected to the L_2_ and L_3_ loops [13]. The interaction of the Zn ion with its ligands was treated through a bonded approach in which the Zn-N and Zn-S bonds and S-Zn-S angles, were set according to the parameters provided in references [14,15,16,17]. The binding of DBD to DNA occurs within L_1_ and L_3_ loops in a region, conventionally chosen to be the northern part of the molecule.

The secondary structure of miR4749 was derived from its single sequence (reported above) by minimizing the free energy folding of an RNA sequence, according to a routine implemented in the webserver RNAFOLD, under default parameters [18]. To predict a 3D structure for miR4749, the secondary structure of miR4749, including the corresponding dot-bracket notation, was submitted to the SIMRNA software [19]. Successively, the obtained 3D best models for miR4749 were submitted to a computational docking with the DBD structure by applying HDOCK, a docking hybrid algorithm combining a template-based modeling and ab initio free docking [20]. The corresponding docking score was used to estimate the protein-ligand binding free energy. All the structure figures were created by Pymol [21] and VMD [22].

### 2.4. Molecular Dynamics (MD) Simulations

MD simulations of DBD, miR4749 and the DBD-miR4749 complex in water were carried out by the GROMACS 2018 package [23], using AMBER03 Force Field for the protein and miR4749 [24], and SPC/E for water [25]. miR4749 was centered in a in a cubic box of edge 7.0 nm, while DBD and the DBD-miR4749 complex were centered in a cubic box of edge 9.0 nm^3^. Simulations were performed by following the procedures described in references [17,26]. Briefly, boxes were filled with water molecules, to reach a hydration level of 9 g water/g protein. The ionization states of protein residues were fixed at pH 7, and Cl^−^ or Na^+^ ions were eventually added to keep the system electrically neutral. H bonds were constrained with the LINCS algorithm [27], while the Particle Mesh Ewald (PME) method [28,29] was used to calculate the electrostatic interactions with a lattice constant of 0.12 nm. Periodic Boundary Conditions in the NPT ensemble with T = 300 K and p = 1 bar, with a time step of 1 fs were used. The temperature was controlled by the Nosé-Hoover thermostat with a coupling time constant τ_T_ = 0.1 ps [30], while the Parrinello–Rahman extended-ensemble, with a time constant t_P_ = 2.0 ps, was used to control pressure [31]. Each system was minimized and then heated to 300 K with steps at 50 K, 100 K, 150 K and 250 K. MD trajectories were analyzed by the GROMACS package tools [23]. Each model was submitted to 10 ns long MD trajectory, replicated three times. The equilibration of each system was checked by analyzing the Root Mean Square Displacement (RMSD) as a function of time.

### 2.5. Calculation of the Binding Free Energy

The binding free energy, ΔG_B_, of the DBD-miR4749 complex, was evaluated by the Molecular Mechanics Poisson–Boltzmann Surface Area (MM-PBSA) method, by following the procedure as reported in references [15,32,33]. Briefly, ΔG_B_, was estimated from: ΔG_B_ = ΔG_complex_ − (ΔG_receptor_ + ΔG_ligand_), with the free energy, G, given by: G = E_MM_ − TS_MM_ + G_solv_, where E_MM_ is the internal energy, TS_MM_ is the entropic term and the G_solv_ the solvation contribution, further decomposed into an electrostatic (G_polar,solv_) and non-polar (G_nonpolar,solv_) parts [34]. The E_MM_ energy was evaluated from E_MM_ = E_elec_ + E_VdW_, where = E_elec_ is the protein–protein electrostatic and E_VdW_ is the Van der Waals interaction energy. The entropic contribution was estimated by the quasi-harmonic approach as reported in reference [35]. G_polar,solv_ was evaluated by numerically solving the Poisson–Boltzmann equation with the Adaptive Poisson–Boltzmann Solver (APBS) software [36], setting a 0.561 × 0.553 × 0.549 Å grid-spacing, and using the AMBER03 force field parameters a probe radius of 1.4 Å for the dielectric boundary. The dielectric constant was set to 2 for the interior and to 87.5 for water [37]. The non polar part of the solvation contribution was evaluated by G_non polar,solv_ = γ SASA + β, with γ = 2.27 kJ mol^−1^nm^−2^ and β = 3.84 kJ/mol [38]. For each model of the complex, an average free energy was evaluated by taking into consideration 20 snapshots, recorded every 0.5 ps from the last 1 ns of the 10 ns long MD simulation runs, for each of the three replicates.

## 3. Results and Discussions

### 3.1. Fluorescence Quenching Results

The capability of DBD to interact with the miR4749 was investigated by fluorescence spectroscopy. Figure 2A shows the fluorescence emission spectrum of DBD excited at 295 nm (black line), at which, the tryptophan residue (Trp146) of DBD still absorbed, while the other aromatic residues (Tyr and Phe) were substantially not excited. The spectrum was peaked at about 346 nm (see the arrow), indicating that Trp146 was almost fully exposed to the solvent, as also was the case in previous works [7,17]. Such a result finds a correspondence with the rather high SASA value (about 80 Å^2^) for Trp146 from the DBD X-ray structure.

A progressive reduction of the DBD fluorescence emission was detected upon adding increasing concentrations of miR4749 (color lines in Figure 2A). Additionally, no significant wavelength shift of the emission peak was observed; this being indicative that the addition of miR4749 did not affect the solvent exposition of Trp146, as also observed for the interaction of DBD with miR-21-3p [7]. It is interesting to note that a similar quenching behavior has been observed for the interaction of DBD with the anticancer peptide p28 [17].

Figure 2B shows the ratio, *F*_0_/*F*, between the fluorescence emission detected at 346 nm, of DBD alone and of DBD in the presence of progressively higher concentrations of the quencher *Q* (here miR4749). The data follow a linear trend and they can be described by the Stern–Volmer equation [9]:
(1)$$F_{0}/F=1+K_{q}\tau_{q}\left[Q\right]=1+K_{SV}\left[Q\right]$$
where *k_q_* is the bimolecular quenching constant, *K_SV_* is the Stern–Volmer quenching constant, and τ_*q*_ is the average lifetime of Trp146 of DBD in the absence of a quencher; with an average lifetime τ_*q*_ of (2.79 ± 0.02) × 10^−9^ s having been measured.

The *K_SV_* constant, extracted from the linear fit of *F*_0_/*F* data by Equation (1) (see black lines in Figure 2B) was found to be (1.68 ± 0.06) × 10^5^ M^−1^, while the corresponding bimolecular quenching constant, *k_q_*, (determined from *k_q_* = *K_SV_*/τ_*q*_) was (6.0 ± 0.3) × 10^13^ M^−1^s^−1^. Notably, the *k_q_* value is much higher than the diffusion-controlled quenching value, which is typically about 10^10^ M^−1^s^−1^, indicating a static quenching mechanism [9]. To further support the static nature of the quenching, we measured the lifetime of Trp146 in the presence of miR4749 at a 1:1 molar ratio. The found value of (2.82 ± 0.02) × 10^−9^ s was almost the same measured for the DBD Trp146 alone (see above). Definitely, a static quenching mechanism can be assumed, and then the formation of a stable complex between DBD and miR4749 in the ground state [9]. Accordingly, the Stern–Volmer constant *K_SV_* represents the affinity constant, *K_A_*, of the complex. The obtained value for *K_A_* of about 10^5^ M^−1^ witnesses the formation of a specific complex between DBD and miR4749 with moderate affinity, similarly to that found for the DBD-miR-21-3p complex [7]. In both cases, the occurrence of a static quenching without any shift of the fluorescence peak, can be put into a relationship to an allosteric interaction mechanism. In other words, the binding of miR4749 to DBD can induce a conformational change on DBD, which, in turn, affects the Trp146 fluorescence.

### 3.2. FRET Results

With the aim to extract structural information on the interaction between DBD and miR4749, we applied FRET, by following an experimental procedure similar to that used for the study of the DBD-miR-21-3p complex [7]. Briefly, the lone Trp146 of DBD constitutes the donor (D), while the Atto390 dye, bound to the 5′ end of miR4749, plays the role of the acceptor (A). We remarked that Trp and Atto390 represent an appropriate *D*–*A* couple since the emission spectrum of Trp146 shows a high overlapping with the absorption spectrum of miR4749Atto390 (not shown), similarly to what was previously reported [7]. Accordingly, the *D*–*A* distance, *R*, can be put into relationship to the energy transfer efficiency (*E_FRET_*) through the expression [9]:(2)$$E_{FRET}=R_{0}^{6}/R_{0}^{6}+R^{6}$$
where the Förster radius, *R*_0_, is the *D*–*A* distance at which *E_FRET_* is 0.5. For the DBD-miR4749Atto390 pair, *R*_0_, calculated through the Förster formula [9], has been found to be *R*_0_ = 2.5 ± 0.1 nm, which is practically the same value obtained for the Trp-Atto390 pair [39].

With the aim to evaluate the *D*–*A* distance, we first determined *E_FRET_*, from the quenching of fluorescence emission of D in the presence of A, through the equation [9]:(3)$$E_{FRET}=1-F_{DA}/F_{D}$$
where *F_D_* and *F_DA_* are the fluorescence emission intensities of *D* alone (DBD-miR4749) and in the presence of *A* (DBD-miR4749Atto390), respectively. Fluorescence emission spectra of DBD-miR4749 (red dashed line) and of DBD-miR4749Atto390 (black solid line), excited at 295 nm are shown in Figure 3A; both the spectra being obtained at a 1:1 molecular ratio. Notably, miR4749 bound to Atto390, induce a higher quenching of the Trp146 fluorescence in comparison to bare miR4749 (see Figure 2); such a behavior being indicative of an energy transfer from *D* to *A*. The fluorescence emission intensities, *F_D_* and *F_DA_*, at 346 nm, determined from ten experiments on independently prepared samples, allowed us to obtain, through Equations (2) and (3), an average EFRET value of (6.9 ± 0.3) × 10^−2^, from which, by Equation (5), an average *D*–*A* distance (R) of 3.8 ± 0.2 nm was derived.

Additionally, we analyzed the enhancement of the fluorescence emission of *A* in the presence of *D*. Figure 3B shows the representative emission spectra of miR-749Atto390 (black curve) and of the DBD-miR4749Atto390 (red dashed line), both of them having been excited at 295 nm. The peak at 460 nm of miR4749Atto390 was found to be enhanced after the addition of DBD (1:1), consistently with an energy transfer from *D* to *A*. Accordingly, *E_FRET_* was evaluated by the following expression [9]:(4)$$E_{FRET}=\left(F_{AD}/F_{A}-1\right)\left(\varepsilon_{A}/\varepsilon_{D}\right)$$
where *F_A_* and *F_AD_* are the fluorescence emission intensities of *A* (miR4749Atto390) and of *D* (DBD-miR4749Atto390), respectively, while ε*_A_* and ε*_D_* are the molar extinction coefficients of *A* (ε*_A_* = 2700 M^−1^cm^−1^) and of *D* (ε*_D_* = 1500 M^−1^cm^−1^) at the exciting wavelength of 295 nm [9,10,40]. Upon measuring *F_A_* and *F_AD_*, the *D*–*A* distance (R) can be determined through Equations (2) and (4). Measurements on ten independently prepared samples allowed us to find out *R* = 4.0 ± 0.2 nm, which is in good agreement with that derived from the donor fluorescence quenching method.

Finally, the *D* lifetime variation method was used to calculate *E_FRET_* through the equation [9]:(5)$$E_{FRET}=1-\left(<\tau_{DA}>/<\tau_{D}>\right)$$
where <τ*_DA_*> and <τ*_D_* > are the average *D* lifetime in the presence and absence of *A*, respectively. A lifetime of (2.82 ± 0.02) × 10^−9^ s was obtained for the donor DBD-miR-4749, while for DBD-miR4749Atto390, (i.e., *D* in the presence of *A*), a lower lifetime (<τ*_DA_*> = (2.68 ± 0.02) × 10^−9^ s) was detected; such a reduction further confirms the occurrence of FRET in the DBD-miR4749Atto390 system. Accordingly, through Equations (2) and (5), we evaluated a *D*–*A* distance of 4.1 ± 0.1 nm, which is slightly higher than the value determined by the two other methods; such a discrepancy can be ascribed to an overestimation of FRET efficiency in the latter cases [10].

### 3.3. Modelling and Docking

FRET experiments allowed us to evaluate the distance between Trp146 of DBD and the dye (Atto390) bound to the 5′ end of miR4749; such a structural parameter can provide a valuable help to predict the interaction regions between miR4749 and DBD. We therefore developed a computational procedure aimed at modeling the structure of the DBD-miR4749 complex (see also Section 2.3). From the sequence of miR4749, we first derived its secondary structure by the RNAFOLD program [18]. Such a structure, shown in Figure 4A together with the corresponding dot-bracket notation, has been used to find out a 3D model for miR4749 through the SIMRNA software [19]. The obtained best five models for miR4749 (labeled from miR4749-M1 to miR4749-M5) are shown in Figure 3A. We noted small differences mainly in correspondence of the tail arrangements.

All these five structures have been separately submitted to a docking procedure with DBD through the HDOCK suite [20]; for each ligand–receptor couple, the first ten ranked models having been taken into consideration for further analysis. The extracted 50 selected most suitable models for the DBD-miR4749 complex have been found to be characterized by a docking energy score ranging from −250 to −180 a.u. A preliminarily visual inspection of these models revealed that the ligand (miR4749) mainly binds at three different regions of the receptor (DBD). In particular, in 22 models, miR4749 binds at the top of DBD around the Zn-finger motif where the DNA-p53 binding mainly occurred (see Figure 1). Additionally, in 12 models, miR4749 binds at the right lateral part of DBD, while in five models, it binds at the bottom of DBD; in the remaining models, miR4749 binding at other portions of DBD.

For each model, we measured the distance, *D_DA_*, between the center of the aromatic rings of the lateral chain of Trp146 belonging to DBD and the 5′ end of miR4749; the corresponding histogram being shown in Figure 5A. Generally, the *D_DA_* distance spanned from 1 to 6 nm, with the largest part of complexes (about 66%) being characterized by a *D_DA_* value between 3 and 4.5 nm. By taking into consideration that the FRET results indicated an experimental *D_DA_* distance within the 3.6–4.2 nm range and by assuming a contribution of 0.1 nm, as due to the dye attached to the 5′ end of miR4749, we preliminarily discarded those models whose *D_DA_* value was lower than 3.7 nm or higher than 4.3 nm. The remaining 16 models were then grouped by evaluating the RMSD between each couple of models by following the procedure as described in reference [17] In particular, models whose RMSD value differed less than 0.1 nm were grouped together, with the first ranked model taken as a representative of the corresponding group.

After that, we obtained five models (labeled as Models 1–5), which are collectively shown in Figure 5B. Notably, in three models, miR4749 was located at the top of DBD, while in the other ones, it was located at the right side opposite to the Trp146. Finally, these five models were submitted to a 10 ns long MD simulation (with three replicates) in order to evaluate the corresponding binding free energy, ΔG_B_ (see Section 2.5).

Figure 6A shows representative RMSDs vs. time for each of the five models. In all the cases, we noted an increasing trend up to about 2 ns, followed by a transient different for the various runs lasting 1–2 ns. Successively, the RMSDs exhibit oscillations around an average value ranging within 0.30–0.45 nm. Similar results were obtained for the other runs, as also it is shown in Figure 6B.

Table 1 reports the resulting ΔG_B_ value, together with the contribution of the different components, the corresponding *D_DA_* distance being also reported (column 2).

For all the five models, the final ΔG_B_ values are negative, indicating energetically favorable bound states. Accordingly, all the five models could represent a plausible structural representation of the DBD-miR4749 complex. Concerning the various components of ΔG_B_, negative contributions are observed for both the solvation terms (ΔG_nonpol solv_ and ΔG_pol solv_), while the other two terms (ΔG_MM_ and −TΔS_MM_) give a positive value. Additionally, ΔG_nonpol solv_ and −TΔS_MM_ values are rather small, while ΔG_pol solv_ and ΔG_MM_ are much higher than the other ones. Notably, the ΔG_pol solv_ term, corresponding to the electrostatic component of ΔG_B_, exhibits the most significant differences among the five models, and, at the same time, it provides the highest contribution to ΔG_B_, with this suggesting an electrostatic guide for the formation of the complex between DBD and miR4749. It is interesting to note that binding free energies have been obtained, in the same range of ours, by a slightly different computational method for the interaction between DBD and DNA [41].

We further remarked that Model 3 was characterized by the lowest ΔG_B_ value. On such a basis, we assumed that it represented the best model for the DBD-miR4749 complex. As shown in Figure 7A, this model was consistent with a binding of miR4749 at the top of DBD, where the L_3_ loop and the Zn-finger motif were located. Notably, such a region is crucial for the p53 functional binding to DNA, as it is evident from Figure 7B showing the X-ray structure of DBD complexed with DNA. This found a correspondence with the high sequence similarity of the miR4749 with the DNA Response Element (RE) of p53 family members [8]. Furthermore, the binding of miR4749 to DBD involves the H_1_ helix of DBD, responsible for the DBD dimerization [42]. Similarly to what is suggested for the interaction between DBD and miR-21-3p [7], it could be hypothesized a possible role played by miR4749 in the impairment of the p53 DNA binding function, as well as in the oligomerization of p53, with both actions being susceptible to inhibit the p53 oncosuppressor function.

## 4. Conclusions

The study of the interaction between miR4749 and the DBD portion of the tumor suppressor p53, by fluorescence spectroscopy put into evidence the formation of a specific complex between DBD and miR4749 with an affinity of about 10^5^ M. Additionally, careful FRET measurements were provided an estimation of an average distance of 3.9 ± 0.3 nm between the intrinsic Trp146 of DBD and the dye bound to miR4749. Such structural information was exploited in a computational approach, suitably developed, to predict a global structure for the complex. The found best model for the DBD-miR4749 complex indicates that miR4749 bound to DBD in correspondence of the DNA binding site. Accordingly, it could be hypothesized that the interaction of miR4749 with p53 could impair the p53 DNA binding function leading to an inhibition of the p53 oncosuppressor function. Such a hypothesis deserves some interest and it should be further tested by in vivo studies in order to deeply address the interplay between p53 and miR4749. However, the direct action of miR4749 on p53 might inspire new therapeutic strategies finalized to restore the p53 anticancer function.

## Figures and Tables

**Figure 1 biomolecules-10-00346-f001:**
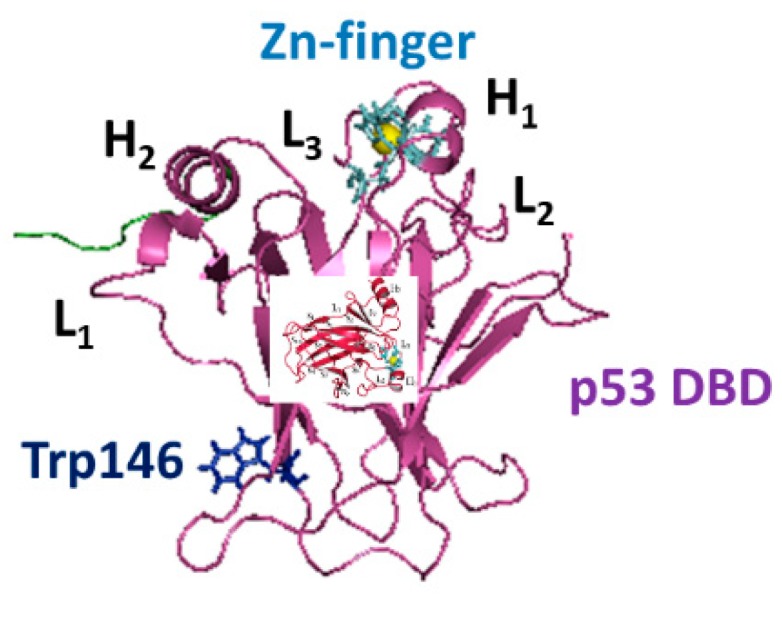
The structure of p53 DNA binding domain (DBD) derived from the chain B of PDB entry 1 TUP, with the addition of the 290–300 AA portion (colored in green). The Zn-finger and Trp146 are shown as sticks, while Zn as a yellow sphere.

**Figure 2 biomolecules-10-00346-f002:**
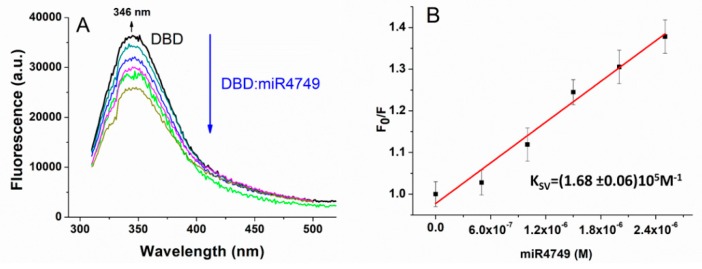
(**A**) Fluorescence emission spectra of DBD (1 µM) alone (black line) and at progressively higher concentrations of miR4749 (0.5-2.5 µM; colored lines) at 298 °K, by exciting at 295 nm and corrected for the Raman scattering of the buffer. (**B**) Stern–Volmer plot of the fluorescence quenching of DBD (1 µM in PBS buffer) as a function of miR4749 concentration at 298 °K, shown as black squares. Continuous red line is the linear fit through Equation (1); the extracted Stern–Volmer constant, K_SV_, being reported.

**Figure 3 biomolecules-10-00346-f003:**
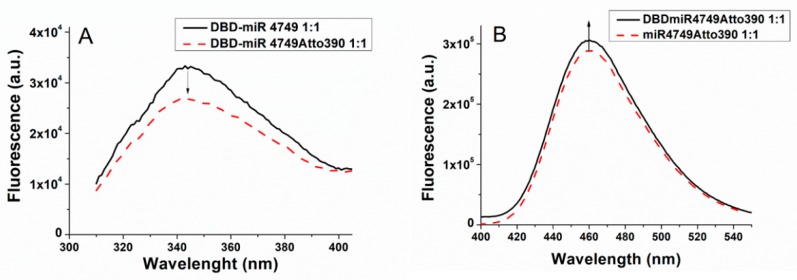
(**A**) Fluorescence emission spectra of DBD-miR4749 (black line) and of DBD-miR4749Atto390 (red dashed line), obtained at a concentration of 1 µM with a 1:1 molar ratio between DBD and miR4749 or miR4749Atto390. (**B**) Fluorescence emission spectra of DBD-miR4749Atto390 (black line) at a concentration of 1 µM and of miR4749Atto390 (red dashed line); both of them at a concentration of 1 µM, with a 1:1 molar ratio. All the spectra were excited at 295 nm and corrected for the Raman scattering of the buffer.

**Figure 4 biomolecules-10-00346-f004:**
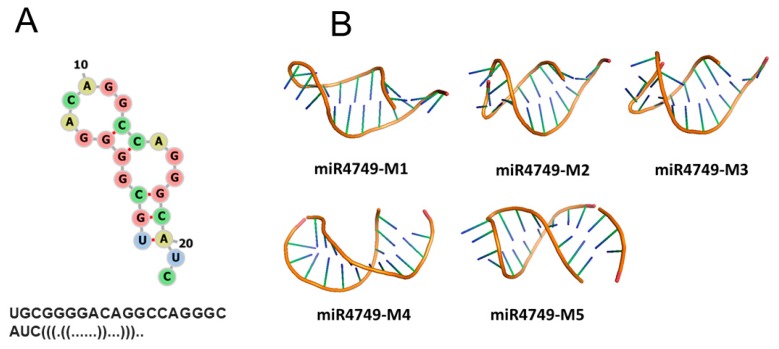
(**A**) Secondary structure of miR4749 together with the dot-bracket representation (bottom). (**B**) Five best models for the 3D structure of miR4749.

**Figure 5 biomolecules-10-00346-f005:**
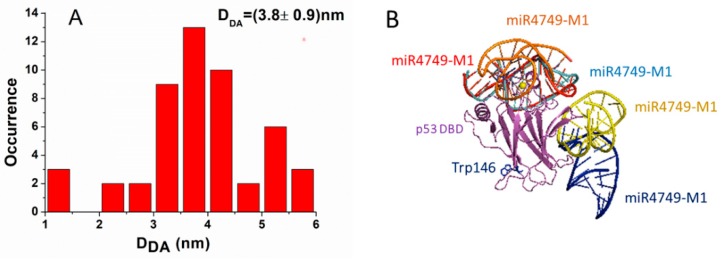
(**A**) Histogram of the *D_DA_* distance in the 50 models for the DBD-miR4749 complex. The distance was measured between the 5′ end of miR4749 to which the Atto390 dye is bound and the center of the aromatic rings of the lateral chain of Trp146 of DBD. (**B**) Collective representation of the five best models for DBD-miR4749 complex. DBD is colored in magenta, while miR4749: Model 1 (red), Model 2 (blue), Model 3 (orange), Model 4 (yellow) and Model 5 (azur).

**Figure 6 biomolecules-10-00346-f006:**
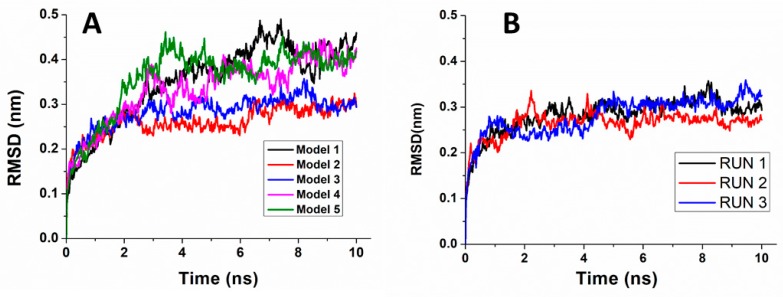
Temporal evolution of the all atom Root Mean Square Displacement (RMSD) for: (**A**) a representative run for each of the five DBD-miR4749 models and (**B**) the three replicate runs of Model 3.

**Figure 7 biomolecules-10-00346-f007:**
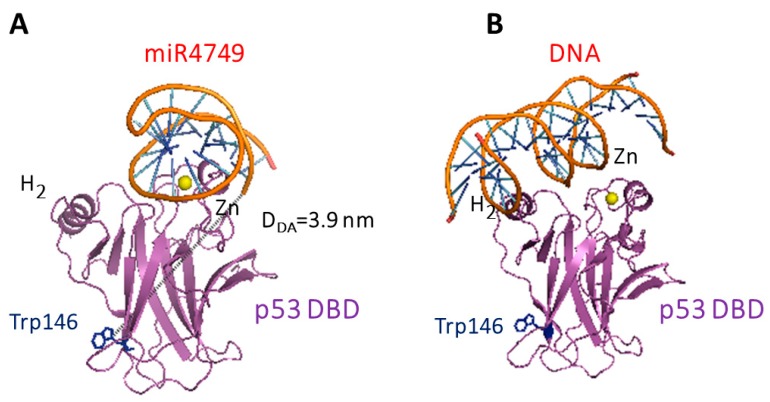
(**A**) Best model for the complex between DBD and miR4749. The distance *D_DA_* between Trp146 and the 5′ end of miR4749 and the center of the aromatic rings of the lateral chain of Trp146 of DBD (black dashed line) is reported. (**B**) X-ray structure of DBD (chain B) complexed with DNA (1 TUP PDB entry).

**Table 1 biomolecules-10-00346-t001:** Binding free energy, ΔG_B_, and its components for the five best DBD-miR4749 interaction models. ΔG_nonpol solv_ represents the nonpolar contribution to the solvation term, ΔE_MM_, the internal energy, -TΔS_MM_, the entropic term, and finally the ΔG_pol solv_ is the electrostatic contribution to the solvation term. *D_DA_* is the distances between the 5′ end of miR4749 and the aromatic ring center of the lateral chain of Trp146.

MODEL	*D_DA_* nm	G_nonpol solv_ kJ/mol	E_MM_ kJ/mol	-TS_MM_ kJ/mol	G_pol solv_ kJ/mol	G_B_ kJ/mol
**Model 1**	4.2	−34	2.74 × 10^4^	536	−6.89 × 10^4^	−4.10 × 10^4^
**Model 2**	4.1	−33	2.73 × 10^4^	535	−4.89 × 10^4^	−2.11 × 10^4^
**Model 3**	3.9	−38	2.76 × 10^4^	537	−8.14 × 10^4^	−5.33 × 10^4^
**Model 4**	3.8	−35	2.78 × 10^4^	541	−7.66 × 10^4^	−4.83 × 10^4^
**Model 5**	4.2	−31	2.76 × 10^4^	543	−5.90 × 10^4^	−3.09 × 10^4^

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
