# Peer review of "Investigation of a Direct Interaction between miR4749 and the Tumor Suppressor p53 by Fluorescence, FRET and Molecular Modeling"

_biomolecules, 2020, doi:10.3390/biom10020346_

Round 1

Reviewer 1 Report

The authors describe their attempt to identify interactions between the DNA binding domain (DBD) of the tumor suppressor p53 and miR4749 with the help of fluorescence spectroscopy combined with computational modelling and docking studies. FRET results showed a distance of (3.9±0.3) nm, between the lone tryptophan of DBD and an acceptor dye suitably bound to miR4749. Authors found that the interaction of miR4749 involves the DBD L3 loop and the H1 helix and miR4749 might be inhibiting the p53 interaction with DNA. However, the study has investigated an important interaction but similar studies for p53 DBD have been already investigated. Though techniques and methods used in the article are very appropriate and relevant but study lack novelty. p53 is very comprehensively studied protein and there are various other known regions by which p53 could be inhibited. Inhibition of p53 functional activity by Mir4749 could be one reason for loss of its function but it cannot be established until other invivo studies are performed. So it could be very naive to say that "could help to develop new therapeutic strategies able to restore the p53 functional activity". 

Author Response

First of all, we would like to thank the Reviewer for his/her comments about the appropriateness of our methods for studying the interaction between DBD and miRNA. Concerning the novelty, we would like to remark that a direct interaction between mature miRNA and DBD is quite unusual and it is still object of extensive debate. Indeed, only two evidences have been provided in  the literature (one of them having been obtained in our lab). In this context, further evidences about a direct binding between DBD and different miRNAs could give insight to understand the miRNAs role in cancer development.  On the other hand, we agree that the p53 loss of functionality may not depend only from the interaction of microRNA with its DNA binding region and that  our results might be complemented by in vivo experiments. However, such a hypothesis is intriguing and it could inspire new molecular strategies. Accordingly, the related sentences have been slightly modified (Pag.1 Lines 23-24, Pag.2 Line 71 and Pag.10 Lines 369-369).

By hoping that the manuscript can now be acceptable for publication  

Sincerely yours

Salvatore Cannistraro

Reviewer 2 Report

Investigation of a direct interaction between miR4749 and the tumor suppressor p53 by Fluorescence, FRET and molecular modelling

This manuscript describes the structural modelling of a complex between a miRNA and the DNA binding domain of p53, guided by information obtained from FRET. The miRNA was found to bind close to where DNA binds on p53, which suggests that the miRNA could inhibit the transcriptional activity of p53.

Comments:

Why was chain B of PDB structure 1TUP used for the modelling when chain A has fewer missing residues?

What were the MD simulation parameters used for the zinc ion and the His and Cys residues bound to it?

I doubt 10 ns is long enough for the p53-RNA complex structures to equilibrate. Please perform longer simulations that are supported by RMSD plots to show that the structures have equilibrated.

No replicates of MD simulations were performed, which should be standard practice by now. Would like to see the average and standard deviation of binding free energies obtained from the replicates.

Magnitude of delta Gb seems too high. Should be on the order of 10^3 kJ/mol. Please check again.

Figure 6 should include the p53-DNA complex structure for comparison

Author Response

Why was chain B of PDB structure 1TUP used for the modelling when chain A has fewer missing residues?

Indeed, chain B of the 1TUP PDB entry was used in our previous works investigating the interaction between DBD and different molecules, such as Azurin, the p28 peptide, miR21 (see refs.[7,15-17]). Accordingly, we have preferred to use such a chain even in the present work, also  in the perspective to compare the  results and then to better elucidate the binding mechanisms. On the other hand, it should be noted that even chain A is also not completed and it includes  only two AA residues more than chain B.

 What were the MD simulation parameters used for the zinc ion and the His and Cys residues bound to it?

The approach used for treating the Zn-ligands is substantially the same as that used in our previous MD simulation studies on DBD (see refs.[7,15-17]), as mentioned in the manuscript.  However, some more details about the Zn-ligands have been now provided (Pag.3 Lines 115-119), as required by the Reviewer.

I doubt 10 ns is long enough for the p53-RNA complex structures to equilibrate. Please perform longer simulations that are supported by RMSD plots to show that the structures have equilibrated.

The choice of limiting the MD runs to 10 ns has been dictated by having checked  that the system has equilibrated during this time interval. This aspect has been mentioned in the text, showing also a representative plot of the RMSD vs. time for each system (Pag.9 Lines 318-321 and new Fig.6A).  

No replicates of MD simulations were performed, which should be standard practice by now. Would like to see the average and standard deviation of binding free energies obtained from the replicates.

As required, three replicates were done for each model.  The RMSDs  vs. time for three replicates (of one model) are now  shown (see new Fig.6B). The corresponding binding free energy values,  as obtained by averaging over  the three replicates,  are  reported in new Table1; a comment having been also added (Pag.9 Lines 321).  We note that variations in the binding free energies values are less than 4% ,   as compared with  the  previous ones.

Magnitude of delta Gb seems too high. Should be on the order of 10^3 kJ/mol. Please check again.

As suggested, we have double-checked the obtained binding free energy (DeltaGb) values  and the same values reported in Table 1 were  obtained (it should be  noted that the values reported in the new Table 1 are  slightly different  as due to the evaluation of replicates). We would like to remark that DeltaGb values are mainly controlled by the electrostatic solvation term (Epol solv) , which is expected to be strongly affected by the involvement of the charged Zn-finger region in the binding with miRNA. In this respect, it should be also mentioned that our DeltaGb values are of the same order (around 104 kJ/mol) of those found for the p53-DNA binding   by a QM-MM study (see ref.Koulgi et al. PLOS ONE 10(11): e0143065). A comment on such an aspect has been added to the manuscript (Pag.9 Line 337-339 and newly added ref.[41]).

Figure 6 should include the p53-DNA complex structure for comparison 

As required, the structure of the the p53-DNA complex has been added to the manuscript (see new Fig.7B) together with a comment (Pag.10 Lines 348-349).

We wish to thank   the Reviewer  for  comments and observations, since we believe that the manuscript has been much improved.

By hoping that the manuscript can now be acceptable for publication  

Sincerely yours

Salvatore Cannistraro

Round 2

Reviewer 2 Report

The authors have adequately addressed my comments.

This manuscript is a resubmission of an earlier submission. The following is a list of the peer review reports and author responses from that submission.